# Differentiable Optimization of Generalized Nondecomposable Functions using Linear Programs

**Zihang Meng[1], Lopamudra Mukherjee[2], Yichao Wu[3], Vikas Singh[1], Sathya N. Ravi[3]**
[1]University of Wisconsin-Madison
[2]University of Wisconsin-Whitewater
[3]University of Illinois at Chicago
`zihangm@cs.wisc.edu, mukherjl@uww.edu, yichaowu@uic.edu`
`vsingh@biostat.wisc.edu, sathya@uic.edu`

## Abstract

We propose a framework which makes it feasible to directly train deep neural networks with respect to popular families of task-specific non-decomposable performance measures such as AUC, multi-class AUC, $F$-measure and others. A feature of the optimization model that emerges from these tasks is that it involves solving a Linear Programs (LP) during training where representations learned by upstream layers characterize the constraints or the feasible set. The constraint matrix is not only large but the constraints are also modified at each iteration. We show how adopting a set of ingenious ideas proposed by Mangasarian for 1-norm SVMs – which advocates for solving LPs with a generalized Newton method – provides a simple and effective solution that can be run on the GPU. In particular, this strategy needs little unrolling, which makes it more efficient during the backward pass. Further, even when the constraint matrix is too large to fit on the GPU memory (say large minibatch settings), we show that running the Newton method in a lower dimensional space yields accurate gradients for training, by utilizing a statistical concept called *sufficient* dimension reduction. While a number of specialized algorithms have been proposed for the models that we describe here, our module turns out to be applicable without any specific adjustments or relaxations. We describe each use case, study its properties and demonstrate the efficacy of the approach over alternatives which use surrogate lower bounds and often, specialized optimization schemes. Frequently, we achieve superior computational behavior and performance improvements on common datasets used in the literature.

## 1 Introduction

Commonly used losses such as cross-entropy used in deep neural network (DNN) models can be expressed as a sum over the per-sample losses incurred by the current estimate of the model. This allows the direct use of mature optimization routines, and is sufficient for many use cases. But in applications ranging from ranking/retrieval to class imbalanced learning, the most suitable losses for the task do not admit a "decompose over samples" form. Examples include Area under the ROC curve (AUC), multi-class variants of AUC, $F$-score, Precision at a fixed recall (P@R) and others. Optimizing such measures in a scalable manner can pose challenges even in the shallow setting.

For AUC maximization, we now know that convex surrogate losses can be used in a linear model Liu et al. [2018], Natole et al. [2018] in the so-called ERM framework. These ideas have been incorporated within deep neural network models and solved using SGD type schemes in Liu et al. [2019]. Such results on stochastic and online data models have also been explored in Ataman et al. [2006], Cortes and Mohri [2004], Gao et al. [2013]. There are also available strategies for measures other

than the AUC: Nan et al. [2012], Dembczynski et al. [2011] give exact algorithms for optimizing $F$-score and Eban et al. [2017], Ravi et al. [2020] proposes scalable methods for non-decomposable objectives which utilizes Lagrange multipliers to construct the proxy objectives. The authors in Mohapatra et al. [2018] discuss using a function that upper bounds (structured) hinge-loss to optimize average precision. Recently, Fathony and Kolter [2020] presented an adversarial prediction formulation for such nondecomposable measures, and showed that it is indeed possible to incorporate such measures within differentiable pipelines.

Our work utilizes the simple observation that a number of these non-decomposable objectives can be expressed in the form of an integer program that can be relaxed to a linear program (LP). Our approach is based on the premise that tackling the LP form of the non-decomposable objective as a module within the DNN, one which permits forward and reverse mode differentiation and can utilize in-built support for specialized GPU hardware in modern libraries such as PyTorch, is desirable. First, as long as a suitable LP formulation for an objective is available, the module may be directly used. Second, based on which scheme is used to solve the LP, one may be able to provide guarantees for the non-decomposable objective based on simple calculations (e.g., number of constraints, primal-dual gap). The current tools do not entirely address all these requirements.

A characteristic of the LPs that arise from the nondecomposable losses mentioned above is that the constraints (including the mini-batch of samples themselves) are modified at each iteration – as a function of the updates to the representations of the data in the upstream layers. In Section 3, we provide LP formulations of widely used nondecomposable terms, which fall squarely within the capabilities of our solver. In Section 4, we show that the modified Newton's algorithm in Mangasarian [2004] can be used for deep neural network (DNN) training in an end-to-end manner without requiring external solvers, where support for GPUs currently remains limited. Specifically, by exploiting self-concordance of the objective, we show that the algorithm can converge globally *without* line search. We then analyze the gradient properties of our approach, and some modifications to improve stability during backpropagation. Our experiments in Section 5 show that this scheme based on Mangasarian's parametric exterior penalty formulation of the primal LP is a computationally effective and scalable strategy to solve LPs with a large number of constraints. On the practical side, we provide two ways to deal with the scaling issue when the constraint matrix is large. On the one hand, we show that sufficient dimension reduction can be used in our solver to solve the problem in a lower dimension space. On the other hand, when the matrix is sparse, with our new sparse implementation, we show that we can train Resnet-18 with $10\times$ mini-batch size (memory savings) on a 2080TI Nvidia GPU. Our code is available, see Fig 1 for an overview of our solver for differentiating many popular nondecomposable objectives.

## 2   Related Works

**Differentiable optimization of nondecomposable objectives.** One closely related result to ours is Liu et al. [2019] which explored optimizing a surrogate loss of AUC using SGD type methods that made the stochastic AUC optimization more practical for deep neural networks. Compared with Liu et al. [2019], our work enables optimizing many more types of nondecomposable objectives for deep neural networks, and relies on the LP reformulation instead of a surrogate loss. In Fathony and Kolter [2020], (the expected values of) such metrics are approximated by LPs defined by predictor and adversary marginal distributions (represented as square matrices). The ADMM solver in Fathony and Kolter [2020], as we will see, inherits some drawbacks of ADMMs including higher memory footprint and slower convergence. Another body of work is the one involving the use of blackbox solvers Pogančić et al. [2019], Berthet et al. [2020] which can be adapted to solve the nondecomposable objectives (e.g., reformulating as LPs). For example, one idea is to use appropriate (infimal or gaussian) convolution to construct approximations of the gradients through the LP. Song et al. [2016] proposes a using a task loss to perturb the loss and is similar in spirit to blackbox solvers – they both solve a sequence of perturbed argmax/argmin problems first, and then use carefully designed procedures for the backward gradient that often require at least one more solve of the optimization problem. We will discuss these properties later in the paper, and in our experiments.

**Differentiable LP solver.** In principle, of course, backpropagating through a convex optimization model (and in particular, LPs) is quite possible. For LPs, we can take derivatives of the optimal value (or the optimal solution) of the model with respect to the LP parameters, and this can be accomplished by calling a powerful external solver. Often, this would involve the overhead of running

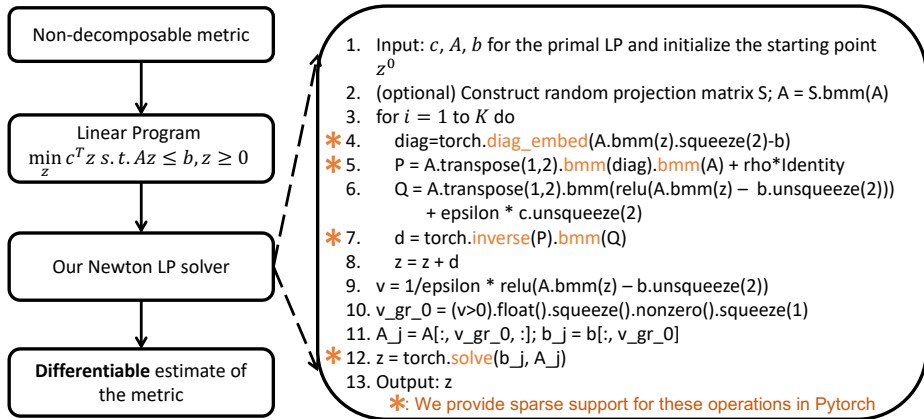

Figure 1: Overview of how to compute differentiable non-decomposable metric and code example of our LP solver. We provide sparse support for operations which are not supported in official Pytorch library (colored operations in the figure).

the solver on the CPU. Solvers within CVXPY, are effective but due to their general-purpose nature, rely on interior point methods. OptNet Amos and Kolter [2017] is quite efficient but designed for quadratic programs (QP): the theoretical results and its efficiency depends on factorizing a matrix in the quadratic term in the objective (which is zero/non-invertible for LPs). The primal-dual properties and implicit differentiation for QPs do not easily translate to LPs due to the polyhedral geometry in LPs. The ideas in Meng et al. [2020] are only applicable when the number of constraints is approximately equal to the number of variables – an invalid assumption for the models that we study here.

# 3 Nondecomposable Functions and corresponding LP models

We first present a standard LP form and then reparameterize several generalized nondecomposable objectives in this way, summarized in a table in the appendix. We start with the binary AUC, extend it to multi-class AUC, and then later, discuss a ratio objective, $F$-score. The appendix also includes a discussion of other objectives that are amenable to our method.

## 3.1 Notations and Generalized LP formulation

**Notations.** We use the following notations:
**(i)** $n$: number of samples used in training.
**(ii)** $X \in \mathbb{R}^{n \times \dim_x}$ : the explanatory features fed to a classifier (e.g., parameterized by $\mathbf{w}$);
**(iii)** $f(x_i)$ (or $f(i)$): a score function for the classifier where $x_i \in X$;
**(iv)** $Y \in \{0, 1\}$: target label and $\hat{Y} \in \{0, 1\}$: predicted label for binary classification, both in $\mathbb{R}^n$;
**(v)** $A \otimes B$: Kronecker product of matrices $A$ and $B$.
**(vi)** $I_r$: Identity matrix of size $r$ and $\mathbb{1}$ is the indicator function.
**(vii)** $\mathbf{B}_{k,-}$ (and $\mathbf{B}_{|,k'}$) gives the $k$-th row (and $k'$-th) column of $\mathbf{B}$.

**LP formulation.** We consider a general linear program (LP) that contains nonnegative variables as well as inequality and equality constraints. The form of the LP is given as

$$\max_{\tilde{u}, \tilde{v}} g^T \tilde{u} + h^T \tilde{v} \quad \text{subject to} \quad E\tilde{u} + F\tilde{v} \leq p, \quad B\tilde{u} + G\tilde{v} = q \quad \tilde{u}, \tilde{v} \geq 0$$

We can write it more succinctly as

Variable $z = [\tilde{u} \ \tilde{v}]$; Coefficient $c = [-g \ -h]$; Constraints $A = \begin{bmatrix} E & B & -B \\ F & G & -G \end{bmatrix}^T$; Constants $b = [p \ q \ -q]^T$

The corresponding primal LP can be written as, $\min_z c^T z$ subject to $Az \leq b, \ z \geq 0$.

## 3.2 Maximizing AUC

The Area under the ROC Curve (AUC) calculates the probability that a classifier $f(\cdot)$ will rank a randomly chosen positive sample *higher* than a randomly chosen negative sample. AUC varies

between $0$ and $1$, where $1$ represents all positives being ranked above the negatives. AUC may be estimated using the Wilcoxon-Mann-Whitney (WMW) Statistic Hanley and McNeil [1982], as

**Definition 3.1** (AUC). Let $n$ be the number of samples. Let $X_+$ (and $X_-$ resp.) be the set of positive (and negative resp.) samples such that $|X_+| + |X_-| = n$ where $|\cdot|$ is the cardinality of the set. Then, AUC is given as $\left(\sum_{i=1}^{|X_+|} \sum_{i=1}^{|X_-|} \mathbb{1}_{f(x_i) > f(x_j)}\right) / \left(|X_+| \, |X_-|\right)$ for $x_i : i \in \{1, \cdots, n\}$.

Here, we follow Ataman et al. [2006] to calculate the AUC based on the WMW statistic as follows.

$$\min_{z_{ij}} \sum_{i=1}^{|X_+|} \sum_{j=1}^{|X_-|} z_{ij} \quad \text{s.t.} \quad f(x_i) - f(x_j) \geq \epsilon - z_{ij} \quad \text{where } x_i \in X_+, x_j \in X_-; \quad z_{ij} \geq 0, \quad (1)$$

where $\epsilon$ is a given constant. Problem (1) computes AUC indirectly by minimizing the number of pairs (one each from the positive and negative classes) where the positive sample is not ranked higher than the negative sample: so, the number of zero entries in $z$ equals the number of pairs where this condition is not true. For a given $z$, we have that,

$$\text{AUC} = \left(n - \|z\|_0\right) / \left(|X_+| \, |X_-|\right) = \left(n - \sum_i \sum_j \epsilon^{-1} \text{relu}(0, -z_{ij} + \epsilon)\right) / \left(|X_+| \, |X_-|\right).$$

If $z_{ij}$ is $0$, then $\epsilon^{-1}\text{relu}(0, -z_{ij} + \epsilon)$ equals $1$. Otherwise $z_{ij} > 0$, it follows from the first constraint in (1), that $z_{ij} \geq \epsilon$, so $\epsilon^{-1}\text{relu}(0, -z_{ij} + \epsilon)$ equals $0$. Observe that in (1), the number of constraints is $|X||X_-|$, which is $O\left(n^2\right)$.

### 3.3 Maximizing Multi-class AUC

An extension of AUC to the multi-class setting, $\text{AUC}_\mu$, is defined in Kleiman and Page [2019]. The $\text{AUC}_\mu$ objective optimizes an indicator matrix calculated on the orientation function, $O_{i,j}$ defined as,

**Definition 3.2** (Orientation Function; Kleiman and Page [2019]). Assume we have $K$ classes $\{y_1, \cdots, y_K\}$. Let $\mathbf{f}(x_i, -) \in \mathbb{R}^K$ (extension of $f(.)$ to the multi-class setting) indicate the model's prediction on $x_i$ for each of the $K$ classes (class-specific probability). Let $x_i^*$ provide the index of $x_i$'s true class label. Let $\mathbf{P} \in \mathbb{R}^{k \times k}$ be a partition matrix where $\mathbf{P}_{k,k'}$ is the cost of classifying a sample as class $k$ when it should be $k'$. Define,

$$\mathbf{v}_{kk'} = \mathbf{P}_{k,-} - \mathbf{P}_{k',-} \text{ and } \widetilde{\mathbf{v}} = \mathbf{v}_{x_i^* x_j^*} \in \mathbb{R}^K; \quad O_{i,j} = (\widetilde{\mathbf{v}}_{x_i^*} - \widetilde{\mathbf{v}}_{x_j^*})(\langle \widetilde{\mathbf{v}}, \mathbf{f}(x_i, -)\rangle - \langle \widetilde{\mathbf{v}}, \mathbf{f}(x_j, -)\rangle).$$

The LP formulation of $\text{AUC}_\mu$ is fully characterized by $\mathbf{P}$. We can set $\mathbf{P}(k, k) = 0 \; \forall k$ and $1$ for all other entries. We can also define a $\mathbf{P}$ with arbitrary entries or formulate AUC in a one-vs-all setting (denoted as $\text{AUC}^{\text{ova}}$). Here, for presentation, use the simplified $0/1$ partition matrix $\mathbf{P}$. Let $\widetilde{\mathbf{f}}(i, j) = \mathbf{f}(x_i, x_i^*) - \mathbf{f}(x_j, x_i^*) + \mathbf{f}(x_j, x_j^*) - \mathbf{f}(x_i, x_i^*)$. Then the LP formulation is given by,

$$\text{AUC}_\mu^{\text{bin}} : \min_{z_{ij}} \sum_{i=1}^n \sum_{j=1:x_i^* < x_j^*}^n z_{ij} \quad \text{s.t.} \quad \widetilde{\mathbf{d}}_{ij} \widetilde{\mathbf{f}}(i, j) \geq \epsilon - z_{ij}, \quad \forall i, j : x_i^* < x_j^* \quad z_{ij} \geq 0 \quad (2)$$

where $\widetilde{\mathbf{d}}_{ij} = \widetilde{\mathbf{v}}_{x_i^*} - \widetilde{\mathbf{v}}_{x_j^*}$. Problem 2 can be seen as an extension of our binary AUC model, where $z_{ij}$ is the ranking between a pair of points defined for multiple classes.

### 3.4 Maximizing $F$-score

The $F$-score (or $F$-measure) is a representative of objectives expressed as ratios of some combination of True positives (TP), False positives (FP), True negatives (TN) and False negatives (FN). We use the result in Dembczynski et al. [2011] to express the $F$-score in the ratio form and further relax it into the LP form. The detailed LP formulation of $F$-score and other ratio functions is in the appendix.

# 4 Backpropagation via Fast Exterior Penalty Optimization

Unlike traditional feedforward networks, where the output of each layer is a relatively simple (albeit non-linear) function of the previous layer, a LP layer must solve a constrained problem, therefore implementing scalable/efficient backpropagation schemes that minimizes overhead requires more care and is an active topic of research. This problem is, of course, not unique to LPs and manifests in differentiable sorting Mena et al. [2018] and formulating quadratic or cone programs Amos et al. [2017]. One may unroll gradient descent steps Amos et al. [2017], Goodfellow et al. [2013], Metz et al. [2017] or use projections Zeng et al. [2019]. Recently Agrawal et al. [2019] introduced a package for differentiable constrained convex programming, which includes LPs as a special case. For LPs, Meng et al. [2020] presents an unrolled scheme and Blondel et al. [2020] shows that we can differentiate through LP formulations of sorting/ranking *exactly* by using smooth approximations of projection steps. Berthet et al. [2020] describes an interesting approach where one computes approximate gradients through ranking/shortest path problems by stochastic perturbation techniques.

**Remark 1.** *Some previous works Zeng et al. [2019] have considered LPs where the constraints are deterministic (for a fixed input dimension), i.e., do not depend on the data $X$, which is different from the LPs in Sections 3.2–3.4.*

Note that perturbation techniques in Berthet et al. [2020] are applicable to our LPs as well. The Fenchel Young losses in Berthet et al. [2020] is attractive because there is no need to compute the Jacobian. Implementation-wise, one could think of the backward pass as a function given the input and output of the forward pass. But the gradient expressions of the losses involves an expectation and hence may require multiple calls to a LP solver in order to approximate the expectation. Parallelization and warm starts were shown to alleviate this dependency to some extent by sampling in parallel.

**Rationale for our approach.** Consider a LP with a large $m$ number of constraints in fixed dimensions $n$ ($n \ll m$). This assumption holds in all formulations in Section 3. This is because we assume that the architecture is fixed whereas minibatch size depends on the complexity of the task (stable gradient or when noise in gradient is high). Hence, solving such LPs using off-the-shelf solvers as in Berthet et al. [2020], in the general case, could slow down training. The strategy in Agrawal et al. [2019] does offer benefits over Amos and Kolter [2017] for sparse QPs. Our strategy is to run Mangasarian's Newton's method on an exterior penalty function of the LP. There are two advantages: **(i)** during **forward** pass, quadratic local convergence of Newton's method indicates that unrolling the method may be reasonable; and further **(ii)** based on the relationship between dual and primal variables, and the exactness of the exterior penalty we can show that **backward** pass is independent of $m$. We will discuss both results and some modifications to deal with discontinuous Hessian (and its inverse) that is required for Newton's method. A similar approach is adopted in Amos and Kolter [2017] where Primal-Dual Interior Point methods with implicit differentiation is used for differentiation purposes. But the exterior penalty in (3) satisfies a nice property: primal and dual solutions are related by a *closed form* expression which can be exploited for efficient backpropagation.

## 4.1 Forward Pass using Newton's Algorithm on a Sufficiently Reduced Space

Fast automatic (forward or reverse mode) differentiation requires performing the *forward* pass efficiently. In our setting, we seek to solve and backpropagate through an LP. We will focus on reverse mode differentiation since it is the most suitable for DNN training.

Given a primal LP, for a fixed accuracy $\varepsilon > 0$, Mangasarian [2004] solves an unconstrained problem,

$$\min_{y} g(y) := \frac{1}{2} \left\| \sigma(Ay - b) \right\|^2 + \varepsilon c^T y, \tag{3}$$

where $\sigma(\cdot) = \max(\cdot, 0)$ represents the elementwise relu function. A modified Newton's method can be used to solve (3) that performs the following iterations:

$$y = y + \lambda d \tag{4}$$

where

$$d = \tilde{H}(y)^{-1} \nabla g(y) := (\nabla^2 g(y) + \rho I)^{-1} \nabla g(y) \tag{5}$$

In large scale settings of $A, b$, such Newton methods are known to perform empirically better than gradient descent Mangasarian [2006], Keerthi et al. [2007]. We will evaluate if this holds for our purposes shortly.

**Why is Newton's method applicable for minibatches?** In general, the convergence of Newton's method depends strongly on initialization (even for convex problems), i.e., we can only provide local convergence results. However, this is not the case for our problems since in our examples, either the level sets are bounded from below; or the feasible set is compact, as noted in Mangasarian [2004]. There are two reasons why the above result, by itself, is insufficient for our purposes: **(i)** it assumes that we can perform line search to satisfy Armijo condition; **(ii)** even with line search, the result does not provide a *rate* of convergence. In DNN training, such line search based convergence results can be very expensive. The key difficulty is handling the discontinuity in the Hessian. As a remedy, we use self concordance of (3) to guarantee global convergence of (4) iterations for the exterior penalty formulation in (3). To do so, we first show a result (proof in appendix) that characterizes the discrepancy between the actual Hessian of (3) and the modified one in (4) when the (feature) matrix $A$ is randomly distributed.

**Lemma 1.** *Assume that $A$ is a random matrix, and fix arbitrary $y \in \mathbb{R}^n$. Then with probability one, $g$ in (3) can be approximated by a quadratic function (given by $\tilde{H}, \nabla g(y)$) over a sufficiently small neighborhood of $y$.*

Intuitively, Lemma 1 states that with probability one, each $y$ has a neighborhood in which the Hessian is constant. In addition, the modified Hessian is nonsingular at all points (in particular the optimal $y^*$), and so we can then show the following global convergence result.

**Theorem 2.** *Assume that the primal LP has a unique optimal solution, and that the level set $\{z : Az \leq b, c^T z \leq \alpha\}$ is bounded for all $\alpha$ (for dual feasibility). Then short step (no line search) Newton's method converges globally at a linear rate with local quadratic convergence.*

*Proof.* First, note that the objective function is piecewise quadratic since it is a sum of piecewise quadratic functions defined by coordinatewise relu function $\sigma$. In particular, $g$ is self concordant since its third derivative is zero almost everywhere. Now setting $\rho < \varepsilon$, we see that an approximate solution of the problem with the modified Hessian is also an approximate solution to equation 3. Moreover, since the possible values of $\tilde{H}$ is finite, the local norm (also known as Newton's decrement) $\nabla g(y)^T \tilde{H}(y)^{-1} \nabla g(y)$ is finite. Hence, we can choose $\rho > 0$ so that there is a descent direction $d$, that is, there exists a step size $\lambda > 0$ such that $\lambda \nabla g(x)^T d < 0$. Finally, we use Theorem 4.1.12 in Nesterov [2013] to claim the desired result. □

The assumptions in Theorem 2 are standard: 1. uniqueness can easily be satisfied by randomly perturbing the cost vector; 2. in most of our formulations, we explicitly have bound constraints on the decision variables, hence level sets are bounded.

**Remark 2.** *Convergence in Thm. 2 is guaranteed under standard constraint qualification assumptions. Linear Independent Constraint Qualification (LICQ) is satisfied for AUC, and Multi-class AUC formulations in §3. But the F-score formulation does not satisfy LICQ, hence we need safeguarding principles in the initial iterations (until iterates get close to the optimal solution).*

Our analysis of the Newton's method for LP so far indicate that we may be able to use a constant step size $\lambda$ (avoid linesearch) to obtain fast convergence provided that we are able to choose $\rho$ sufficiently small. For our purposes, we assume $\lambda$ to be a hyperparameter and can be tuned by cross validation.

**Sketching $d$ using Sufficient Dimension Reduction (SDR).** For training the backbone network, we have to compute $d$ in (4) using $A$ (and $b$) which could be very large within each training iteration. But each minibatch corresponds to an LP instantiation in (3), so it may suffice to compute an approximate $d$ in each Newton's iteration. To approximate $d$, we use sufficient dimension reduction in which a lower dimensional LP obtained by projecting $A$ is solved Zhang et al. [2020], Kim et al. [2020]. That is, instead of the inverse of $\tilde{H}$, we will compute a lower dimensional sketch of $\tilde{H}$ by using $SA$ (and $Sb$) instead of $A$ (and $b$) for some sampling matrix $S$ and solve problem (3). During each iteration, the metrics are computed on the current minibatch as is done in Fathony and Kolter [2020]. Thus (without SDR) the memory cost is directly proportional to minibatch size viz., number of samples and feature dimensions. *The advantage of using SDR (as opposed to naive sketching) is that the lower dimensional space can also be chosen using data driven cross validation or other techniques*

*Kim et al. [2016], making it ideal for training purposes.* In essence, the size of minibatch does not matter – as long as the minibatch can be sufficiently reduced, our solver is directly applicable, especially for low resource, memory constrained settings. For a fixed (low) dimension parameter, our sketched $\underline{d}$ is,

$$\underline{d} = \underline{H}^{-1}\underline{g} \quad \text{where} \quad \underline{g} = \frac{\partial}{\partial y}(\|\text{relu}(Ay - b)\|_2^2) \approx (SA)^T S \cdot \text{relu}(\text{sign}(Ay - b)), \tag{6}$$

$$\underline{H} \approx (\text{diag}(S \cdot \text{relu}(\text{sign}(Ay - b)) \odot S \cdot \text{relu}(\text{sign}(Ay - b))) \cdot SA)^T \cdot SA. \tag{7}$$

It is easy to see that the approximate update can be seen as equivalent to Iterative Hessian Sketch algorithm which has geometric convergence rate Pilanci and Wainwright [2016].

## 4.2 Backward Pass using Optimal Dual Variables Aided by Unrolling

The advantage of optimizing the exterior penalty in (3) is that given an iterate $y_t$, accuracy $\varepsilon$, we can get the optimal dual solution $v_t$ by simple thresholding, i.e., $v_t = 1/\varepsilon(A^T\sigma(Ay - b))$. By complementarity slackness, nonzero coordinates of $v_t$ specify the set of active constraints in $Az \leq b$. So, given an approximate $y_t$ such that $\nabla g(y_t)\tilde{H}(y_t)^{-1}\nabla g(y_t) \leq \varepsilon$, to get the primal solution $z^*$, we solve the "active" linear system given by $\tilde{A}\tilde{b}$, where $\tilde{A}$ denotes the active rows of $A$ and the corresponding subvector $\tilde{b}$. So, backpropagation through the layer reduces to computing derivatives of $\tilde{A}^{-1}\tilde{b}$ (simple via automatic differentiation). The appendix gives a systematic way of choosing $\varepsilon$.

**How to backpropagate through unrolled iterations?** We assume that the chain rule is applicable up to this LP layer and is $\frac{\partial L}{\partial z}\frac{\partial z}{\partial A}$ (for one of the parameters $A$), and note that it is possible to find $\frac{\partial L}{\partial z}$ (either directly or using a chain rule). Therefore we focus our attention on $\frac{\partial z}{\partial A}$, which involves the LP layer. Indeed, unrolling each iteration in (4) is equivalent to a "sublayer". So, in order to backpropagate we have to show the partial derivatives of each operation or step wrt to the LP parameters $A$, $b$, and $c$.

Our goal is to calculate $\frac{\partial d}{\partial A}$ where $d = Q^{-1}q$, $Q = \tilde{H}$ and $w = \nabla g(y)$. We can use the product rule to arrive at: $\partial d = -\left(w^T Q^{-1} \otimes Q^{-1}\right)\partial Q + Q^{-1}\partial w$. To see this, note that we have used the chain rule to differentiate through the inverse in the first term. The second term is easy to compute similar to the computation of Hessian. For each of these terms we eventually have to compute $\frac{\partial Q}{\partial \text{var}}$ or $\frac{\partial w}{\partial \text{var}}$ where var $\in \{c, A, b\}$ which can also be done by another application of chain rule. Please see appendix for empirical verification of unrolled gradient and the one provided by $\tilde{A}^{-1}\tilde{b}$.

Before proceeding, we should note an issue that comes up when differentiating each step of the unrolled algorithm because the Hessian is piecewise linear (constant) as a function of the input to that particular layer. Here, some possible numerical approximations are needed, as we describe below.

**Remark 3.** *Note that the diagonal matrix term in $\frac{\partial Q}{\partial A}$ is nondifferentiable due to the presence of the step function. However, the step function is a piecewise constant function, and hence has zero derivative almost surely, that is, in any bounded set $S$, $z \in S$, if a ball (of radius $r > 0$,) $B_r(z) \subseteq S$, then the Lebesgue measure of the set of nondifferentiable points on $S$ is zero. Please see appendix for a formal justification where we show this by approximating the step function using a sequence of logistic functions with increasing slope parameter at the origin.*

Therefore, in this setting, Remark 3 provides a way to compute an approximate sub-gradient when using Newtons method based LP layers. The function is a piece-wise quadratic function and differentiable everywhere, and the inverse of the Hessian acts as a preconditioner.

*Summary.* Our forward pass involved three steps: **(1)** finite steps of Newton's method using which we **(2)** computed the dual variable by a thresholding operation, and **(3)** finally, to get the primal solution, these dual variables are first used to identify the active constraints followed by solving a linear system. To backpropagate through these three steps, we must differentiate through each layer of our procedure including $\tilde{A}^{-1}\tilde{b}$: this is independent of whether we use unrolling or Danskin's theorem. For instance, using Danskin's theorem here would involve differentiating through the fixed point of the Newton's iterations similar to (regularized) gradient descent iterations in iMAML Rajeswaran et al. [2019].

# 5 Experiments

In this section, we conduct experiments on commonly used benchmarks to show that our framework can be used to optimize multiple different objectives within deep neural networks and lead to performance gain. We start with binary AUC optimization, and then extend to multi class AUC optimization and $F$-score optimization. Nonnegative matrix factorization is discussed in the appendix.

**Optimizing Binary AUC**

We follow the Liu et al. [2019] to conduct experiments on optimizing AUC score directly with deep neural networks. The baseline algorithms we compare with for binary AUC are cross-entropy loss and two algorithms (PPD-SG and PPD-AdaGrad) from Liu et al. [2019].

**Datasets:** Cat&Dog, CIFAR10, CIFAR100, and STL10 (See appendix for dataset details). We follow Liu et al. [2019] to use 19k/1k, 45k/5k, 45k/5k, 4k/1k training/validation split on Cat&Dog, CIFAR10, CIFAR100, STL10 respectively.

**Experimental setting.** We follow Liu et al. [2019] to construct an imbalanced binary classification task by using half of the classes as positive class and another half as negative class, and dropping samples from negative class by a certain ratio, which is reflected by the positive ratio (ratio of majority class to the minority class) in Table 1. Con-

Table 1: Binary AUC optimization results on four benchmark datasets.

| AUC(%) | **Cat&Dog** | | | | **CIFAR10** | | | |
|---|---|---|---|---|---|---|---|---|
| **Positive Ratio** | 91% | 83% | 71% | 50% | 91% | 83% | 71% | 50% |
| Cross-Entropy | 67.6 | 74.6 | 85.1 | 87.4 | 65.2 | 73.3 | 78.1 | **83.7** |
| PPD-SG | **79.1** | **81.5** | 85.5 | 87.1 | 69.8 | 73.9 | **79.1** | 82.6 |
| PPD-AdaGrad | 77.3 | 80.6 | 83.7 | 86.3 | 69.7 | 74.1 | 78.4 | 83.1 |
| Ours | 78.6 | 81.3 | **85.6** | **87.8** | **72.5** | **74.4** | 78.3 | 82.7 |

| AUC(%) | **CIFAR100** | | | | **STL10** | | | |
|---|---|---|---|---|---|---|---|---|
| **Positive Ratio** | 91% | 83% | 71% | 50% | 91% | 83% | 71% | 50% |
| Cross-Entropy | 57.8 | 58.4 | 62.2 | 66.3 | 63.5 | 67.1 | 72.7 | 80.8 |
| PPD-SG | 56.5 | 58.9 | 61.6 | 65.2 | **70.7** | 71.6 | 75.1 | 77.4 |
| PPD-AdaGrad | 56.2 | 59.0 | 62.6 | 67.6 | 68.5 | **72.4** | 76.7 | 78.5 |
| Ours | **58.2** | **60.5** | **64.5** | **69.0** | 68.4 | 71.1 | **76.7** | **81.6** |

structing imbalanced datasets by dropping samples was also used in Cui et al. [2019] to construct long tailed CIFAR-LT. We use the same random seed, learning rate and total number of iterations in all of our experiments including multi class AUC and $F$-score experiments. See appendix for model architecture, learning rate, etc.

**Results.** The results are shown in Table 1. We can see that our method slightly outperforms Liu et al. [2019] and outperforms cross-entropy loss by a large margin, especially on imbalanced datasets, where the AUC objective shows superiority over cross-entropy loss.

**Per-iteration complexity.** The wall clock run time per iteration of cross-entropy, our solver, PPD-SG and PPD-AdaGrad are 0.018, 0.069, 0.122 and 0.130 respectively (in terms of seconds). Our method and PPD cost more than directly optimizing cross-entropy because of the additional cost of solving AUC and our method is more time efficient compared with PPD.

**Influence of $\epsilon$ in our formulation 1.** We test the influence of $\epsilon$ using Cat&Dog as an example. Results (see appendix for the table) show that $\epsilon = 0.1$ gets slightly worse performance than $\epsilon = 0.01$, while $\epsilon = 0.001$ performs much worse. To choose $\epsilon$, we follow the approach in Mangasarian [2004]. If for two successive values of $\epsilon_1 > \epsilon_2$, the value of the $\epsilon$ perturbed quadratic function is the same, then it is the least 2-norm solution of the dual. Therefore, we simply choose an $\epsilon$ that satisfies this property, which is chosen to be $0.01$ in our experiments.

**Optimizing Multiclass AUC**

We also evaluate our method for multiclass AUC. Similar to binary AUC, we construct imbalanced multiclass datasets by dividing datasets into 3 classes and drop samples from 2 of them and report one-versus-all AUC (as $\text{AUC}^{\text{ova}}$) and $\text{AUC}^{\text{bin}}_{\mu}$ score

Table 2: Multiclass AUC results on STL10 and CIFAR100. Drop rate is the proportion used when dropping samples from 2 (of 3) classes.

| $\text{AUC}^{\text{ova}}$(%) | **CIFAR100** | | | | **STL10** | | | |
|---|---|---|---|---|---|---|---|---|
| **Drop rate** | 90% | 80% | 60% | 0% | 90% | 80% | 60% | 0% |
| Cross-Entropy | 54.3 | **59.4** | 62.7 | 63.5 | 66.9 | 68.0 | 74.8 | 81.0 |
| Ours | **58.4** | 59.2 | **64.1** | **65.7** | **72.9** | **72.5** | **75.7** | **82.7** |

| $\text{AUC}^{\text{bin}}_{\mu}$(%) | **CIFAR100** | | | | **STL10** | | | |
|---|---|---|---|---|---|---|---|---|
| **Drop rate** | 90% | 80% | 60% | 0% | 90% | 80% | 60% | 0% |
| Cross-Entropy | 55.1 | 60.6 | 65.0 | 64.0 | 68.9 | 69.6 | 75.8 | 82.2 |
| Ours | **60.1** | **61.2** | **66.0** | **67.2** | **76.1** | **74.4** | **77.7** | **84.5** |

. For STL10, we group class $0 - 2, 3 - 5, 6 - 9$ into the three classes, and drop samples from the

first two classes. For CIFR100, we group class $0 - 32, 33 - 65, 66 - 99$ into three classes, and also drop samples from the first two classes.

**Results.** Results are in Table 2. In addition to one-versus-all AUC metric, we also report the performance in terms of $AUC_\mu$ Kleiman and Page [2019] which is specifically designed for measuring multiclass AUC and preserves nice properties of binary AUC such as being insensitive to class skew. Our method outperforms cross-entropy loss on all four datasets and under all different skewed ratios.

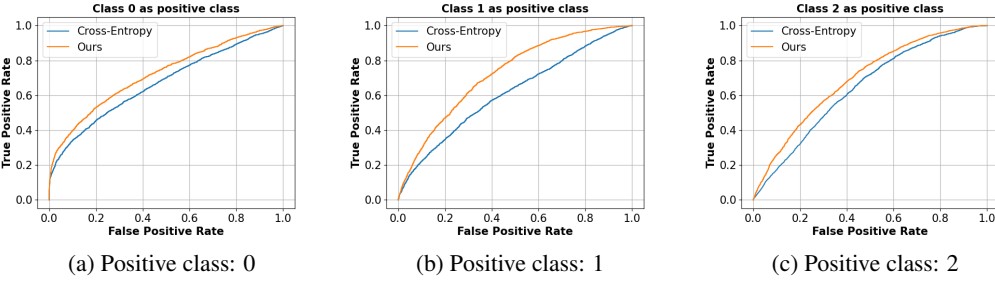

| (a) Positive class: 0 | (b) Positive class: 1 | (c) Positive class: 2 |

Figure 2: ROC curve of multiclass AUC optimization on STL10 with $90\%$ drop rate. We divide STL10 into 3 classes and use one as positive class and other two as negative class to plot the ROC.

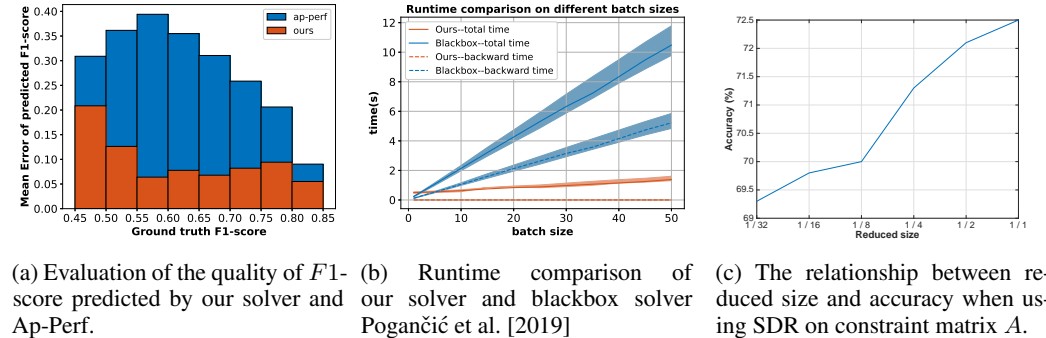

(a) Evaluation of the quality of $F1$-score predicted by our solver and Ap-Perf.

(b) Runtime comparison of our solver and blackbox solver Pogančić et al. [2019]

(c) The relationship between reduced size and accuracy when using SDR on constraint matrix $A$.

Figure 3: Three numerical experiments to show the properties of our solver.

**Optimizing $F1$-score**

We show that by directly optimizing $F1$-score, we can achieve a better performance than when using cross entropy loss. In addition to cross entropy loss, we perform evaluations with two other methods that can also directly optimize the $F1$-score. First, we replace our solver with CVXPY-SCS Agrawal et al. [2019], which refers to the SCS solver used within CVXPY package; second, we perform comparisons with AP-Perf Fathony and Kolter [2020] which offers differentiable optimization of $F1$-score using an adversarial prediction framework. The datasets and our setup to group them into two classes remain the same as in binary AUC section. The results show that our method yields consistent improvement over cross-entropy loss in terms of $F1$-score. The accuracy of cross-entropy loss on Cat&Dog, CIFAR10, CIFAR100 and STL10 is $76.0\%, 70.3\%, 60.4\%, 71.8\%$, respectively, while our method gives $\mathbf{77.8\%}, \mathbf{72.6\%}, \mathbf{63.4\%}, \mathbf{72.7\%}$. When we optimize $F1$-score directly, there exists a local optimal point where assigning all examples to the positive class leads to $F1$-score of $66.7\%$. Both CVXPY-SCS and AP-Perf fall into this local optimal on four datasets (except that CVXPY-SCS gets out of the local optimal point on Cat&Dog and gives $70.1\%$ accuracy). Note that although our solver, CVXPY-SCS and AP-Perf seek to solve the same objective, the backward gradient will be different due to different approximations used by the methods.

**Comparing with blackbox solvers.** Recently, several results Pogančić et al. [2019], Berthet et al. [2020] use blackbox solvers for solving combinatorial problems, which can also be used as a LP solver for problems where we need $dz^*/dc$ ($c$ is the cost vector in standard form LP). These methods can utilize existing LP solvers and compute backward gradient by calling the LP solvers on perturbed LP problems. To our knowledge, the current state-of-the-art LP solvers are often CPU based, which means that one needs to transfer the LP parameters from GPU to CPU, then solve the LP on the CPU before getting the solution back. These CPU-based solvers will usually be slower than our GPU based solver, especially when the batch size is large. To evaluate this property, we randomly

Table 3: Properties of different methods. $c, A, b$ are the parameters describing a linear programming: $\min c^T z, s.t. Az \leq b$. $z^*$ denotes the optimal solution. The extra cost for computing backward gradient for Berthet et al. [2020], depending on implementation, can either be $N$ times time cost, or $N$ times memory cost plus some time cost brought by increased batch size (see appendix for explanation).

| | Ours | CVXPY-SCS | Perturbed solver Berthet et al. [2020] |
|---|---|---|---|
| Gradients support | $\frac{dz^*}{dc}, \frac{dz^*}{dA}, \frac{dz^*}{db}$ | $\frac{dz^*}{dc}, \frac{dz^*}{dA}, \frac{dz^*}{db}$ | $\frac{dz^*}{dc}$ |
| GPU support | ✓ | ✗ | depends on solver |
| Sparsity support | ✓ | ✗ | depends on solver |
| Extra cost for computing backward gradient (denote forward time cost as 1) | Nearly zero | around 1 | $N$ (# perturbations) |

construct LP problems using AUC as an example and make the cost vector the parameter to optimize. In Fig. 3b, we compare the runtime of our solver and the blackbox solver in Pogančić et al. [2019] (with the LP solver from scipy package). We can see that our GPU based solver is not sensitive to the increase in the batch size, while the CPU based solver is, due to the lack of support for mini-batch operations. Our solver has a more clear advantage in backward pass because the matrix needed for computing gradient is already computed during forward pass thus the backward pass is nearly **free**, while the blackbox solver Pogančić et al. [2019] needs at least the same time as forward pass to compute backward gradient. We also compare our solver with Berthet et al. [2020] in Table 3.

**Dealing with large scale LPs.** In practice we often encounter large scale LPs whose constraint matrix may be too large to fit into the GPU memory. We first show that we can utilize sufficient dimension reduction in Fig. 3c, which demonstrates a reasonable tradeoff between reduced size and accuracy. Then we show that when the constraint matrix is sparse, we can readily utilize sparse operations to save memory. We have incorporated functionality to use sparsity within our solver flaport [2020], rusty1s el al. [2020]. Consider AUC maximization in which the constraint matrix is extremely sparse. With our sparse implementation, we are able to run SGD on Resnet-18 with minibatch size of up to 200 on a 2080Ti GPU with just 1GB memory whereas the same problem can take upto 11GB using dense operations ($\approx 10\times$ memory savings) with some overhead.

**Limitations.** Scaling is still a key limitation for differentiable LP solvers. We present two ways to mitigate these problems but there is an associated cost. Our proposed SDR sacrifices some accuracy. Partly due to support within available libraries sparse operations can sometimes be slow when the size of the matrices is large which should improve with better sparsity support in existing libraries.

# 6   Conclusions

We demonstrated that various non-decomposable objectives can be optimized within deep neural networks in a differentiable way under the same general framework of LPs using a modified Newton's algorithm proposed by Mangasarian. A number of recent papers have studied the general problem of backpropagating through convex optimization modules, and this literature provides several effective approaches although scalability remains a topic of active research. Our work complements these results and shows that the operations needed can be implemented to utilize the capabilities of modern deep learning libraries. While our experimental results suggest that promising results on binary AUC, multi-class AUC and $F$-score optimization within DNNs is achievable, we believe that the module may have other applications where the number of constraints are large and data-dependent. Our code is available at https://github.com/zihangm/nondecomposable-as-lp.

**Potential negative societal impacts.** Optimizing non-decomposable measures such as AUC is a fundamental task in statistics/machine learning and not tied to specific applications, so we do not see a negative societal impact of the technical development described here. It is possible, however, that modules such as these can be used within general applications that are detrimental to society.

# 7   Acknowledgments

This work was supported by NIH RF1AG059312 and RF1AG062336, and NSF CCF #1918211. We thank Sameer Agarwal for Twitter discussions regarding implicit differentiation in deep networks, and later pointing to Olvi's paper over email correspondence. Sathya Ravi was supported by UIC start-up funds. Yichao Wu was supported by NSF grant DMS-1821171.

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
