# Differentiable Optimization of Generalized Nondecomposable Functions using Linear Programs: Supplementary Material

**Zihang Meng**[1], **Lopamudra Mukherjee**[2], **Yichao Wu**[3], **Vikas Singh**[1], **Sathya N. Ravi**[3]
[1]University of Wisconsin-Madison
[2]University of Wisconsin-Whitewater
[3]University University of Illinois at Chicago
zihangm@cs.wisc.edu, mukherjl@uww.edu, yichaowu@uic.edu
vsingh@biostat.wisc.edu, sathya@uic.edu

## 1 Examples of solving non-decomposable objectives by LP

### 1.1 LP formulation for Integral Metrics such as Multi-Class AUC

Area Under Curve (for instance, of the Receiver Operating Characteristic curve) provides a graphical summary of the performance of a binary classifier as the classification threshold is varied. The most simplest approach to extend binary AUC to multi-class is by considering multiple one-versus-all pairs which can be formulated as a pairwise separable optimization problems (each) given by:

$$
\text{AUC} : \min_{z_{ij}} \sum_{i=1}^{n} \sum_{j=1:x_i^* \neq x_j^*}^{n} z_{ij} \tag{1}
$$
$$
\text{s.t.} \quad (\mathbf{f}(x_i, x_i^*) - \mathbf{f}(x_j, x_i^*)) \geq \epsilon - z_{ij} \forall i, j : x_i^* \neq x_j^*,
$$
$$
z_{ij} \geq 0
$$

It is well known that such pairwise extensions have many technical caveats including basic properties such as consistency of the estimator. Recently, Kleiman and Page [2019] used the classical U-Statistic to extend the definition to multiclass settings that alleviates such common pitfalls. In our multi-class AUC experiment, we use this one-versus-all AUC as training loss and report performance in both one-versus-all AUC and $\text{AUC}_\mu^{\text{bin}}$. In addition, we can also consider the setting of $\text{AUC}_\mu$ where $\mathbf{P}$ is set arbitarily. In this case, the exact terms in orientation function $O$ proposed by Kleiman and Page [2019] can be written as follows:

$$
\text{AUC}_\mu^{\text{arbit}} : \min_{z_{ij}} \sum_{i=1}^{n} \sum_{j=1:x_i^* < x_j^*}^{n} z_{ij} \tag{2}
$$
$$
\text{s.t.} \quad \widetilde{\mathbf{d}}_{ij} \sum_{k=1}^{K} \widetilde{\mathbf{v}}(k)(\mathbf{f}(x_i, k) - \mathbf{f}(x_j, k)) \geq \epsilon - z_{ij}
$$
$$
\forall i, j : x_i^* < x_j^*,
$$
$$
z_{ij} \geq 0
$$

Note that $\text{AUC}_\mu^{\text{arbit}}$ has the same number of constraints and variables as $\text{AUC}_\mu^{\text{bin}}$. Once the LPs are solved, the loss function is calculated the same way as binary AUC.

35th Conference on Neural Information Processing Systems (NeurIPS 2021), virtual.

## 1.2  LPs for Ratio Objectives and Latent Probabilistic Modeling

Often times, we are interested in assessing the classification performance of submodules in (larger) networks. In such settings, we have optimize general *probabilistic* loss functions, especially when used such loss functions are used in hidden layers. The key property in such formulations is that the overall training loss function contains ratio terms that are unsuitable for training purposes. In this section, we study a subset of non-decomposable metrics, which are typically expressed as ratios of some combination of True Positives (TP), False Positives (FP), True Negatives (TN) and False Negatives (FN). These can be expressed in a general form as

$$\frac{a_{11}TP + a_{12}}{a_{21}TP + a_{22}FP + a_{23}FN + a_{24}} \tag{3}$$

where $a_{pq}$ are constants/cofficients which if set to 0, means the term is absent and not equal to zero in other cases. This formulation can used to define Fscore, $F_\beta$, Jaccard, IOU and Precision at fixed recall. In the following section, we describe the formulation of Fscore as a representative of this approach, other metrics can be formulated similarly.

Given $Y$ the ground truth, our goal is to compute $\hat{Y}$ both of length $n$, which aligns with $Y$ based on the specific metric. We first show how to write TP, FP, TN and FN wrt to these vectors.

$$TP = Y^T \times \hat{Y} \qquad FP = (1 - Y)^T \times \hat{Y}$$
$$TN = (1 - Y)^T \times (1 - \hat{Y}) \quad FN = (Y)^T \times (1 - \hat{Y})$$

### 1.2.1  LP Formulation of $F$-score

The $F$-score (or $F$-measure) is a representative of objectives expressed as ratios of some combination of True Positives (TP), False Positives (FP), True Negatives (TN) and False Negatives (FN). The general form of the ratio functions and formulations for other objectives is in the supplement. Specifically, $F$-score is defined as follows:

**Definition 1.1** ($F$-score). $F\text{-score} = \frac{2 \times (\text{Precision} \times \text{Recall})}{\text{Precision} + \text{Recall}} = \frac{2\text{TP}}{2\text{TP} + \text{FP} + \text{FN}} = 2(Y^T \times \hat{Y})/(\mathbf{1}^T Y + \mathbf{1}^T \hat{Y})$.

The second equality in the definition is due to a simplification of the precision $\left(\frac{\text{TP}}{\text{TP} + \text{FP}}\right)$ and recall $\left(\frac{\text{TP}}{\text{TP} + \text{FN}}\right)$ expressions based on Dembczynski et al. [2011]. The last part is obtained by replacing TP with $Y^T \times \hat{Y}$, FP with $(1 - Y)^T \times \hat{Y}$ and FN with $(Y)^T \times (1 - \hat{Y})$ as functions of $Y$ and $\hat{Y}$. This leads to the following integer fractional optimization model,

$$F\text{-score} = \max_{\hat{Y}} \frac{\mathbf{c}^T \hat{Y}}{d^T \hat{Y} + b} \tag{4}$$

$$\text{s.t.} \quad \hat{Y}_i \in [0, 1], \ i = 1, \ldots, n \text{ where } \mathbf{c} = 2Y \ d = \mathbf{1} \text{ and } b = \sum_{i=1}^{n} Y_i.$$

To solve this, we first relax the constraint on $\hat{Y}$ and reformulate the model as the following LP, by introducing two variables $z \in R^n$ and $t \in R^1$ where $z = \frac{b\hat{Y}}{\mathbf{1}^T \hat{Y} + b}$, $t = \frac{b}{\mathbf{1}^T \hat{Y} + b}$ and $i \in \{1, \cdots, n\}$:

$$\max_{z,t} \frac{\mathbf{c}^T z}{b} \quad \text{s.t} \quad \underbrace{\mathbf{1}^T z + bt = b}_{(a)}; \ \underbrace{z_i \leq t}_{(b)}; \tag{5}$$

$$\underbrace{\phi(f(x_i))t \leq z_i \leq (1 + \phi(f(x_i)))t}_{(c)}; \ 1 \geq z_i, t \geq 0$$

**Remark 1.** *In our formulation above, (a) ensures the appropriate relation between $z$, $t$ and $\hat{Y}$ and corresponds to (a reformulaton of) the ratio objective as a linear function with a fixed denominator. $\hat{Y}$ is recovered from the solution to the LP by computing $\hat{Y}_i = \frac{z_i}{t}$ when $t > 0$ and $\hat{Y}_i = z_i$ otherwise; (b) sets an upper bound for $\hat{Y}_i \leq 1$; and (c) ties the output of the previous layer $\phi(f(x_i))$ (a classifier*

*score for $\hat{Y}_i$) as a input in this layer. Assume $\phi(.) \in \{-1, 1\}$ (ensured if $\phi$ is sigmoid or tanh) is the indicator of the class label (based on sign), then the constraints become $\hat{Y}_i \geq \phi(f(x_i))$ and $\hat{Y}_i \leq 1 + \phi(f(x_i))$. If $\hat{Y}_i$ is $\{0, 1\}$, the constraints ensures that $\hat{Y}_i = 0$ when $\phi(f(x_i)) \leq 0$ and $\hat{Y}_i = 1$ when $\phi(f(x_i)) > 0$. We obtain (c) after relaxing $\hat{Y}_i$ and replacing it with $z$ and $t$.*

This model imposes $4n$ constraints for $n$ samples. Since this is a maximization, a solution to the LP, $O_{\hat{Y}}$, is an upper bound on the integer objective $opt^*$, and serves as the loss.

**Maximizing Jaccard Coefficient.** The Jaccard Coefficient and Dice Index lead to similar formulation as $F$-score. The Jaccard coefficient can be expressed as:

$$
\begin{aligned}
\text{Jacc}(Y, \hat{Y}) &= \frac{TP}{TP + FP + FN} \\
&= \frac{(Y^T \times \hat{Y})}{\sum_{i=1}^n y_i + \sum_{i=1}^n \hat{y}_i - \sum_{i=1}^n y_i \times \hat{y}_i} \\
&= \frac{(Y^T \times \hat{Y})}{\mathbf{1}^T Y + (1 - Y)^T \hat{Y}}
\end{aligned}
\tag{6}
$$

This can be equivalently written as a linear factional program as shown in equation 4 where $c = Y$, $d = (1 - Y)$ and $b = \mathbf{1}^T Y$. The rest of the construction is similar to $F$-score.

**Maximizing $F_\beta$.** Note that $F_\beta$ which is defined as

$$
F_\beta(Y, \hat{Y}) = (1 + \beta^2) \frac{P(Y, \hat{Y}) \times R(Y, \hat{Y})}{\beta^2 P(Y, \hat{Y}) + R(Y, \hat{Y})}
\tag{7}
$$

where $\beta$ is a user specified parameter (balancing the importance of precision and recall) also permits a similar formulation. Here we simply set $c = (1 + \beta^2)Y$, $d = \mathbf{1}$ and $b = \beta^2 \mathbf{1}^T Y$.

**Maximizing $P@R$.** We begin by defining the maximum precision at fixed minimum recall problem as

$$
\begin{aligned}
P@R\alpha &= \text{maximize} \quad P \quad \text{s.t.} R \geq \alpha \\
&= \text{maximize} \quad \frac{(Y^T \hat{Y})}{\mathbf{1}^T \hat{Y}} \quad \text{s.t.} \quad Y^T \hat{Y} \geq \alpha \mathbf{1}^T Y
\end{aligned}
\tag{8}
$$

This is again a linear fractional objective with a linear constraint. So we can write it as an equivalent Linear program using the same transformation where $c = Y$, $d = \mathbf{1}$ and $b = 0$. $R@P$ on the other hand directly leads to a linear program.

## 2 Additional Experiments

### 2.1 Experimental details of AUC and $F$-score experiments

**Datasets.** Cat&Dog is a dataset from Kaggle which contains 25000 images of cats and dogs. $80\%$ of the dataset is used as training set and the rest $20\%$ as test set. STL10 is inspired by the CIFAR-10 dataset but with some modifications. Each class in STL10 has fewer labeled training examples than in CIFAR-10. We follow Liu et al. [2019] to use 19k/1k, 45k/5k, 45k/5k, 4k/1k training/validation split on Cat&Dog, CIFAR10, CIFAR100, STL10 respectively.

**Construction of imbalanced datasets.** We construct the imbalanced binary classification task by using half of the classes as the positive class and another half as the negative class, and dropping samples from negative class by a certain ratio, which is reflected by the positive ratio (the ratio of the majority class to the minority class).

**Implementation Details.** We use a Resnet-18 He et al. [2016] as the deep neural network for all algorithms. During optimization, the batch size is set to 64. The initial learning rate is tuned in $\{0.1, 0.01, 0.001\}$ and decays $2/3$ at $2k, 10k, 25k$-th iteration. We train $40k$ iterations in total. The $\epsilon$ in Newton's method is $0.001$. We use the same random seed, learning rate and total number of iterations in all of our experiments including multi class AUC and $F$-score experiments. During the

| Objective | AUC | AUC$_\mu^{\text{bin}}$ | $F$-score |
|---|---|---|---|
| $g$ | $\mathbf{1} \in \mathbb{Z}^{\lvert T \rvert \times \lvert N \rvert}$ | $\mathbf{1}^\dagger$ | $\mathbf{c}$ |
| $h$ | - | - | $0$ |
| $E$ | $\mathbf{-1} \in \mathbb{Z}^{\lvert T \rvert \times \lvert N \rvert}$ | $\mathbf{-1}^\dagger$ | $\begin{bmatrix} 1 \\ -1 \\ 1 \end{bmatrix}^{\S}$ |
| $F$ | - | - | $\begin{bmatrix} -1 \\ \phi(f(x_i)) \\ -(1+\phi(f(x_i))) \end{bmatrix}^{\S}$ |
| $p$ | $p_{ij} = (f(x_i) - f(x_j) - \epsilon)$ | $p_{ij} = (\mathbf{f}_{ij}^{\ddagger} - \epsilon)$ | $0$ |
| $B$ | - | - | $\mathbf{d}$ |
| $G$ | - | - | $b$ |
| $q$ | - | - | $b$ |

Table 1: Table showing the general LP coefficients for each model. $\dagger$: length based on problem setting; $\ddagger$: $\mathbf{f}_{ij}^{\ddagger} = \widetilde{\mathbf{d}}_{ij}(f(x_i, y_{C(x_i)}) - f(x_j, y_{C(x_i)}) + f(x_j, y_{C(x_j)}) - f(x_i, y_{C(x_j)}))$; $\S$: one block for each $i \in [1, ..n]$. We do not include NMF in this table, as its formulation is known in the literature Recht et al. [2012].

experiment, We use the same random seed, learning rate and total number of iterations in all of our experiments including multi class AUC and $F$-score experiments.

**Code.** We provide an example code to show how to compute $F$-score differentiably using our solver, CVXPY-SCS, and Ap-Perf respectively in "example_code_Fscore_optimization.ipynb".

## 2.2 Explanation regarding time/memory cost of Perturbed solver relative to our algorithm

When implementing the Perturbed solver from Berthet et al. [2020] to compute the backward gradient, there are two options one can use. The implementation from the authors of Berthet et al. [2020] approaches the problem as follows. Suppose that the number of perturbations used is $M$. During the forward pass, suppose that originally the mini-batch size was $B$. The implementation perturbs the mini-batch $M$ times to form a mini-batch of size $M * B$ and calls the solver (any state-of-the-art solver can be used) to solve this mini-batch. Then this result will be used to calculate the backward gradient. In this way, the extra cost for backward gradient can be seen as the cost of solving a batch of size $M * B$ minus the cost of solving a batch of size $B$ – indeed, since/if we do not need the backward gradient, we can just call the solver to solve this $B$ sized batch. It is easy to see that the extra memory one would need is $M$ times the original memory cost, and the extra time cost will depend on the specific solver (can be $1 - -M$ times the original forward time cost). Another way to implement the above procedure is as follows. During the forward pass, we only solve the $B$ sized batch and during the backward pass, we solve the $B$ sized batch repeatedly $M$ times. In this way, the extra memory cost for the backward pass is the same as the forward one, but the extra time cost will be $M$ times more than the forward pass. In other words, the two options described above point to a trade-off: the first option has an extra memory cost ($M$ times although $M$ can be set to be small) and relatively small extra time cost, while the second option has a low extra memory cost by sacrificing runtime ($M$ times). The reader can verify that in comparison, our method only needs to solve a batch of size $B$ **once** during the forward pass, and the backward pass is almost free. Thereby, our strategy is more efficient compared to both obvious ways of implementing Berthet et al. [2020].

## 2.3 Verification of Unrolling gradient and the one provided by solving at fixed point

We use the $F$-score formulation as an example. For input sample $x$, the neural network predicts a score $f(x)$, and then the scores of a batch of samples will be used to solve the LP form of $F$-score which serves to construct the loss function. We compute the gradient from the final loss function back to the predicted scores from the neural network and compare two possible approaches. One option is that we use $z = \tilde{A}^{-1}\tilde{b}$ as the solution (the one we used/reported in our experiments) where we can compute the gradient with only one step. Another option is that we directly use $y_t$ resulting from the Newton iterations as the solution and compute gradients by unrolling those iterations. We

then compute the cosine value between these two gradient vectors. On experiments on 100 randomly sampled batches, we find that the average cosine value is 0.9991, which means that the two gradients are highly consistent.

## 2.4 How to choose $\epsilon$?

If we can successfully retrieve the active constraints at the optimal solution, we do not need to store the intermediate iterates $y_t$ at all during the forward pass (memory efficient). However, setting $\epsilon$ correctly may be tricky for arbitrary polyhedra since it depends on the geometric properties such as facets and vertices that may be difficult to enumerate. One standard way to get around this is to use a "burn-in" period in which we increase $\epsilon$ slightly in each iteration (of deep network training) and backpropagate through the unrolled Newton's iterations during this period. Once we see that the convergence profile has stabilized, we can fix $\epsilon$ at that value and start using the complementarity conditions and derive the active linear system $\tilde{A}^{-1}\tilde{b}$ as discussed above.

Table 2: Ablation study of $\epsilon$ on Cat&Dog dataset.

| Positive Ratio | 91% | 83% | 71% | 50% |
|---|---|---|---|---|
| Ours($\epsilon = 0.1$) | 71.3 | 77.0 | 84.4 | 87.3 |
| Ours($\epsilon = 0.01$) | 78.6 | 81.3 | 85.6 | 87.8 |
| Ours($\epsilon = 0.001$) | 65.9 | 71.3 | 71.8 | 76.1 |

## 2.5 Experimental results on Nonnegative Matrix Factorization

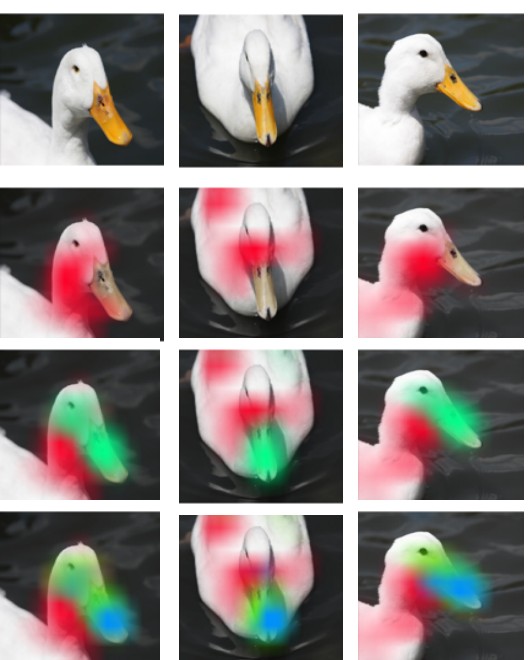

Figure 1: NMF example. Three rows correspond to original images, $k = 1$, $k = 2$ and $k = 3$ respectively.

We demonstrate applicability of our strategy to nonnegative matrix factorization (NMF) by performing a rank $k$ factorization on Convolutional Neural Network (CNN) activations as an example, following Collins et al. [2018]. Recall that the activation tensor of an image at some layer in CNN has the shape $V \in R^{c \times h \times w}$ where $h, w$ are the spatial sizes and $c$ is the number of channels. We can reshape it into $V \in R^{c \times (h \cdot w)}$ and calculate a rank $k$ NMF for $V$: $V = FW$. Each row $W_j$ of the resultant $W \in \mathbb{R}^{k \times (h \cdot w)}$ can be reshaped into a heat map of dimension $h \times w$ which highlights regions in the image that correspond to the factor $W_j$. We show an example for $k = 1, 2, 3$ in Fig.

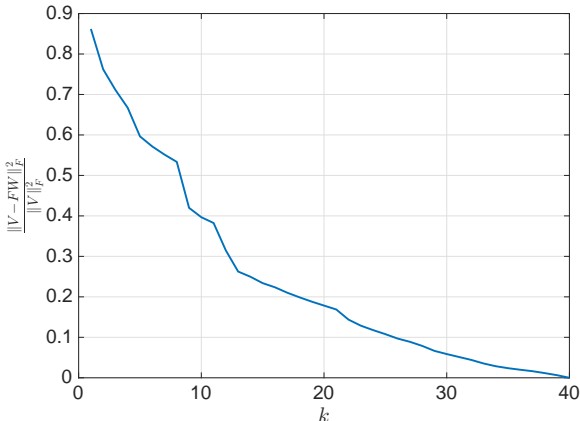

Figure 2: NMF reconstruction error using the sparse implementation of our solver, where $c = 40$, $h \cdot w = 960$.

1. We can see that heatmap consistently captures a meaningful part/concept in the examples. Note that the memory consumption increases quickly with $c$ here since the constraint matrix in our LP formulation is of size $O(c^2) \times O(c^2)$. We utilize the sparse implementation of our solver so that it can fit into GPU memory. See Fig. 2 for the relationship between reconstruction error and the rank $k$, which also shows that our solver can solve NMF problem correctly. Since our method provides backward gradients for the NMF operation, the heatmap generated here can, in fact, be used to construct a loss function during training in order to learn interpretable models.

## 3   Proofs and Details of Results in Section 3

In this section, we will provide the missing proofs and additional calculations in Section 3.

### 3.1   Proof of Lemma 1.

Lemma 1 is restated here for convenience.

**Lemma 1.** *Assume that $A \in \mathbb{R}^{m \times n}$ is a random matrix, and fix some $y \in \mathbb{R}^n$. Then with probability one, $g$ in equation (3) is quadratic (given by $\tilde{H}, \nabla g(y)$) over a sufficiently small neighborhood of $y$.*

*Proof.* Using the integral form of second order Taylor's expansion of $\sigma^2(y) = (\max(0, y))^2$, we can show that,

$$g(y + h) - g(y) - h^T \nabla g(y) = \frac{1}{2} h^T A^T \text{diag}(\mathfrak{d}) Ah \tag{9}$$

where

$$\mathfrak{d} = \int_0^1 \left( \int_0^1 (\sigma(Ay - b))_* ds \right) 2dt. \tag{10}$$

See Remark 1 in Golikov and Kaporin [2019] for details. Without loss of generality, we can assume $b = 0$ by simply translating the origin. Following the same remark, the diagonal matrix coincides with the step function based diagonal in $\tilde{H}$ under the following condition on $h$:

$$e_j^T Ah \cdot e_j^T (Ay) < 0 \implies |e_j^T Ah| \le |e_j^T (Ay)|. \tag{11}$$

Since $y$ is fixed, assuming that the entries of $A$ are chosen from a continuous distribution such that $e_j^T A$ is uniformly distributed over the sphere, then $(e_j^T Ah)^2$ follows a Beta $\left(\frac{1}{2}, \frac{n-1}{2}\right)$ when $h$ is drawn uniformly at random from the unit sphere, independent of $A$. This means that no matter what $y$ is, there exists sufficiently small $h$ such that the left hand side of equation 11 is false with probability one, and in that neighborhood $\text{diag}(\mathfrak{d}) = \tilde{H}$. $\qquad\square$

## 3.2 Differentiating the Step Function in Remark 4

We will use a slightly modified "suffix" notation as in Brookes [2005] in our calculations. That is, for a matrix $A$, $\vec{A}$ is the same as $\text{vec}(A)$, vectorization of $A$ obtained by concatenating all the columns. The following three properties relating the Kronecker product, $\vec{\cdot}$, and differentials will be used often:

1. Fact 1: For two vectors $a, b$, $a \otimes b = \overrightarrow{ba^T}$.

2. Fact 2: If $A$ is $p \times q$ matrix, and $B$ is a $m \times n$ matrix, then $\overrightarrow{\partial B} = (\partial B / \partial A) \overrightarrow{\partial A}$ where $\partial B / \partial A$ is the $(mn) \times (pq)$ Jacobian matrix of $\vec{B}$ with respect to $\vec{A}$. If $A$ or $B$ is a column vector or scalar, then $\vec{\cdot}$ has no effect.

3. Fact 3: $\overrightarrow{\partial(AXB)} = \left(B^T \otimes A\right) \overrightarrow{\partial X}$.

Using the above two facts, we can compute all the gradients needed to backpropagate through the unrolled iterations. We will show the computation for the gradient of $Q^{-1}u$ with respect to $A \in \mathbb{R}^{m \times n}$ for a fixed $u \in \mathbb{R}^n$. We can apply chain rule to the following composition:

$$
\begin{array}{ccc}
A & \xrightarrow{\ f_1 \circ f_2\ } & \left(A^T \tilde{H} A + \rho I\right)^{-1} u \\
{\scriptstyle f_1} \downarrow & {\scriptstyle f_2} \nearrow & \\
A^T \tilde{H} A + \rho I & &
\end{array}
$$

to get, $J_{f_2 \circ f_1} = J_{f_2} \circ J_{f_1}$. Now using Fact 2 on $\partial\left(X^{-1}\right) = -X^{-1}(\partial X)X^{-1}$ and some algebraic manipulation, we obtain,

$$
\overrightarrow{J_{f_2 \circ f_1}} = -\left(u^T \left(A^T \tilde{H} A + \rho I\right)^{-1} \otimes \left(A^T \tilde{H} A + \rho I\right)\right) \overrightarrow{J_{f_1}}. \tag{12}
$$

We will now compute $\overrightarrow{J_{f_1}}$. Note that $\tilde{H}$ is also a function of $A$, so using product rule, we can write $\overrightarrow{J_{f_1}}$ as a sum of three derivatives – with respect to each of $A, A^T, \tilde{H}$. The derivatives with respect to $A$ and $A^T$ are fairly straightforward to compute, so will focus on computing the derivative with respect to $\tilde{H}$. To that end, we will use Fact 3, and show to compute the derivative of the step function by approximating it using the logistic function.

$$
\frac{\partial}{\partial A} \text{diag}\left((Ay - b)_*\right) \approx
$$
$$
\frac{\partial}{\partial A} \text{diag}\left(1 \oslash \left(1 + \exp\left(\kappa\left(-Ay + b\right)\right)\right)\right), \kappa > 0. \tag{13}
$$

Note that these derivatives are used in computing derivatives of upstream network, so using distributional derivatives, and another application of chain rule to the left hand side of equation (18) results in the dirac delta function which is atomic, that is, has all its mass in a measure zero set. Hence this calculation provides an mathematical justification that the set of nondifferentiable points has measure zero for our training purposes. It is easy to formally verify this argument using differentiable tent functions as approximations to the heaviside step function.