# OpenReview forum: "Differentiable Optimization of Generalized Nondecomposable Functions using Linear Programs"
_NeurIPS.cc/2021/Conference — NeurIPS 2021 Poster_

### Official Review · Reviewer_Gi3P · 2021-07-11

**Rating:** 6
**Confidence:** 2

**Summary:**

This paper proposes a new method for training neural networks using LP-based loss functions that involves adapting Mangasarian’s Exterior-point Newton Method. The authors prove new theoretical properties for the Exterior-point Newton method, propose a backward pass technique, and perform experiments on several image data sets to validate their method.


**Limitations And Societal Impact:**

This paper discusses the limitations of the work in detail.

**Main Review:**

**After rebuttal**

I thank the authors for their detailed rebuttal. The authors have clarified my main questions regarding the correctness of the Lemma and using PPD in multi-class AUC. I am happy to recommend acceptance.


However, I strongly urge the authors to revise the clarity of the paper with (i) a proper related works section and (ii) clear and correct statements of the theorems and their proofs. The work has a lot of moving parts and can be difficult to read. As another reviewer mentioned, the paper would benefit greatly with some extra work on improving the clarity.


**Original review**

The experiments in this paper seem convincing that the authors have built a Pytorch library that allows for efficiently maximizing AUC and other LP-type objectives. However, I found overall this paper very difficult to read. It uses ideas from and compares with various works but there is no structure to guide an unfamiliar reader. Trivial topics are explained in detail but core prerequisites are missing. Furthermore the math is difficult to follow: (i) the notation frequently changes and terms are not properly defined, (ii) statements are vague with hidden assumptions. All of this makes it difficult to understand the originality and quality of the contributions of this work.


Related literature: this is presented sporadically in Section 1 and then Section 3.0. In both cases, a reader cannot appreciate this related works because the proposed methods of the paper have not been introduced. I would recommend for clarity that the authors produce a separate Related literature section where they can properly compare the weaknesses of existing methods.

Lines 84-96: The notation and LP setup are useless because the authors do not ever use most of this afterwards in the paper (e.g., target label $Y$ and non-linear function $\phi$). The breakdown of the LP from $c^T x$ to $g^T u + h^T v$ is never used afterwards. This makes the paper even more confusing because notation $x$, $y$, $g$, $u$, etc. are all recycled in Section 3.
- Please try to maintain some consistent notation throughout the paper. In Section 2.1, $x$ is first a row of the data matrix (line 87), then a variable of an optimization problem (line 95). Then in Section 2.2, the variable is now $z$. Then in Section 3, the variable is $y$.

Line 174: Please unpack this math. $\tilde{H}$ is implicitly defined and only here but it is needed later below. Please also define $\rho$.

The statement of Lemma 1 is vague.
- Under what distributional assumptions is $A$ is a random matrix? The answer is hidden in the supplement at the end of the proof, where the authors state their assumption that the rows of $A$ are uniformly distributed over a sphere.
- Typo: “constant” -> “quadratic” ?

The randomness assumption in Lemma 1 leads to the follow up question: is this assumption satisfied in practice? From my understanding of S2.2, the constraint matrices in the LPs contain rows which are just random identity vectors, which is not a continuous uniform distribution over the sphere.

Similarly, the statement of Theorem 2 in the paper is vague. The statement in the supplement is different from the one in the paper but actually includes the core assumptions needed to make the statement meaningful.

The experiments are overall satisfactory although I would have liked to see some more baselines. For example, Multiclass AUC in Table 2 compares only against Cross-Entropy, which is obviously going to be a poor baseline.
- Why can’t you apply PPD to multi-class AUC? If this is a limitation of existing methods then it would have been nice to clarify and make your method more appealing.

It seems to me that the data sets used in numerical evaluation are customized to appear as long-tail distributions. As a future direction, I would be interested to see whether the method in this work could be applied directly to long-tail data sets (e.g., CIFAR10-LT, CIFAR100-LT).

**Time Spent Reviewing:**

5

---

> ### Author Response · Authors · 2021-08-10
> **Author response for Reviewer Gi3P**
>
> Thank you for the time spent in reviewing our paper! We carefully address all your concerns and questions below.
>
> Q1:  Recommend a separate related work section
>
> A1: The reason we did not use a related work section is that we cover most highly related works in our experiment section and the differences among these different approaches appeared to be easier to explain and the paper flowed better. But based on the suggestion, we are happy to add back a related work section summarizing these papers.
>
> Q2: Notations.
> (a) Lines 84-96: The notation and LP setup are useless (target label Y is not used in the following text)
>
> (b) The breakdown from $c^Tx$ to $g^Tu+h^Tv$ is not used afterwards
>
> (c) line 87 uses $x$ as a row of data and line 95 uses $x$ as a variable
>
> (d) Line 174: Please unpack this math to be an equation
>
> (e) $x,y,g,u$ are recycled in section 3
>
> A2: To summarize our answer below, all of our choices of notations were meant to make it easier to read. To do so, we decided to follow the use from [30] which our work builds upon and adjust/modify only when necessary. This led to a little bit of notational overload when adjusting to different types of objectives we covered. Combined with an unfortunate typo, we understand why it may have been confusing. We explain below.
>
> (a) We use Lines 84-96 to introduce the notations, and also implicitly describe all important variables that will be used. Although the following equations do not explicitly utilize the target label $Y$, we felt it was an important variable to orient/introduce for the reader to understand the problem setup.
>
> (b) Our intention is to start from the most general LP form, which includes both the inequality and equality constraints, and then move to the form with only the inequality constraints. Further, this general form is explicitly used in Table 1 in the supplement where we show the general LP coefficients for the several problems studied by us. Without the form noted by the reviewer in comment (b), the blocks in the constraint matrix will not be meaningful.
>
> (c) We use $x$ as a data point (a row of the data matrix $X$) throughout the paper. The only reason we used $x$ as the variable in line 95 is that “min $c^Tx$, s.t. $Ax \leq b$, $x \geq 0$” would likely look familiar for most readers (e.g., also used in [30]) and we want to make sure this is easy to understand at the start of our formulation. This was a regrettable oversight and we will change $x$ in line 95. Thank you.
>
> (d) We will unpack the expression to make it easier to read.
>
> (e) As mentioned above, $x$ refers to a row of data matrix (a data sample throughout the paper except in line 95). We did not use $y$ (lower case) before section 3. We did not recycle $g$. The reviewer will see that in both section 2 and 3 we use g as a functional on the variable (in section 2 it is linear and in section 3 it is nonlinear). The reviewer is correct that we recycled $u$ in section 3. We thank the reviewer for pointing this out and this will be fixed.
>
> Q3: (a) Statements about Lemma 1 and Theorem 2 are vague.
>
> (b) The randomness assumption in Lemma 1 leads to the follow up question: is this assumption satisfied in practice? From my understanding of S2.2, the constraint matrices in the LPs contain rows which are just random identity vectors, which is not a continuous uniform distribution over the sphere.
>
> (c) The statement of Theorem 2 in the supplement is different from the one in the paper but actually includes the core assumptions needed to make the statement meaningful.
>
> (d) Typo: “constant” -> “quadratic” ?
>
> A3: (a) Indeed, we give utmost importance to the clarity of technical statements and so if the reviewer is suggesting that we should include ALL details, we are happy to do so. While Lemma 1 follows standard conventions in most textbooks, Theorem 2 was stated informally with the full version in the supplement to keep the discussion succinct. We are happy to modify to the full version, please see comment (c) below.
>
> (b) Our result in Lemma 1 applies to all problems in Sec 2, and is a much more general statement than what is minimally required for 2.2. There are two ways to apply our result in Lemma 1 for problems in 2.2: (i) we may simply add a small Gaussian perturbation as a preprocessing step with sufficiently small variance, and (ii) as pointed out in the review, the constraint matrix (defining the LP) is a deterministic unit diagonal matrix (with all ones column for epsilon), and so the sign condition (equation 11 in supplement) is satisfied for a sufficiently small positive lambda -- a hyperparameter in our experiments. The reviewer will now immediately see that the version of Lemma that we presented in the main paper is applicable to much more general settings such as NMF, and hence beneficial to the community. We will clarify this point.
>
> (c) We had hoped that the proof sketch will provide the key intuition without notational/assumption clutter since we believe that these are fairly standard in analyzing the Newton’s method (none of the assumptions in our analysis are unique/non-standard). We will definitely include the details from the supplement, as suggested in the review, and Golikov and Kaporin 2019 reference in the statement of Theorem 1 in the main paper.
>
> (d) Yes this should be quadratic. Thanks for pointing this out.
>
>
> Q4: Why can’t you apply PPD to multi-class AUC?
>
> A4: Theoretically it is possible but extending PPD to the multi-class does not appear to be trivial. Observe that the algorithm given in the PPD paper [19] is designed specifically for binary two-class problems. In [19] the authors only briefly mention that theoretically it is possible to extend PPD to multi-class but did not give any concrete algorithm. No experiments on multi-class problems are included. This was the reason why we could not use PPD as a baseline in our multi-class AUC experiments.
>
> Q5: As a future direction, I would be interested to see whether the method in this work could be applied directly to long-tail data sets (e.g., CIFAR10-LT, CIFAR100-LT)
>
> A5: CIFAR10-LT was created by randomly deleting negative samples, and in our experiment the way to create the long-tailed dataset is the same. We can easily add support for CIFAR10-LT and CIFAR100-LT in our codebase when it is released.
>
> We hope that these clarifications will lead to a more favorable evaluation from you. Please let us know if there are any other questions we can answer.

---

> ### Author Response · Authors · 2021-09-01
> **Thanks to Reviewer Gi3P**
>
> We thank Reviewer Gi3P for reading our response and updating the review&score. We will modify our paper to have better clarity and add a more thorough related work section (as mentioned in our response).

---

### Official Review · Reviewer_mCRV · 2021-07-14

**Rating:** 7
**Confidence:** 4

**Summary:**

This work proposes to relax some commonly used discrete metrics into a Linear Program (LP). By using the Newton LP method, the algorithm is able to backpropagate through the LP and thus to train a neural network end-to-end while directly optimizing for the metric. In particular, this submission describes how the AUC, multi-class AUC and F-score can be casted as LPs, and how the resulting problems can be solved with a Newton LP method that can be differentiated through. Experiments are presented on Cat&Dog, CIFAR-10, CIFAR-100 and STL.

**Limitations And Societal Impact:**

Sufficiently discussed.

**Main Review:**

### Strengths
1. The problem is well motivated: end-to-end training for metrics such as AUC would be of practical interest to the machine learning community.
2. The formulation seems sound and amenable to efficient (and differentiable) optimization methods.
3. The submission is evidently the result of a solid amount of work to scale the method for deep learning, including implementations of sparse operations for a common deep learning framework.
4. The experiments (which are "medium-scale" by deep learning standards) show consistent improvements over the baselines.

### Weaknesses
1. This work is difficult to follow and sometimes unclear, and would benefit from extra work on clarity before sharing with the community, hence why I am not giving a higher rating (more details in Questions and Minor comments).
2. The experimental evaluation does not seem to provide a timing comparison with the baselines (e.g. Cross-Entropy): the improved performance should also be put in perspective with the amount of time that each method takes.

### Questions
1. Why is it stated that the number of constraints $m$ is much larger than the number of variables? It seems to me that in e.g. 2.2 there are about as many variables as constraints.
2. Why is epsilon needed in the formulation of 2.2? If the computation of the AUC outputs only binary values, then could not one just use the indicator function $z_{ij} > 1$ (and "backpropagate" through this operation with 0/1 according to whether the value is 0/1)?
3. Is it correct to say that at each iteration, the metrics are only computed on the current mini-batch? (this should be made clearer in the paper). If so, how does the batch-size affect the performance of the algorithm, since the metrics are not decomposable over the samples?
4. L.177 the paragraph title is "Why is Newton’s method applicable for minibatches?" but the content does not seem to address this question, can this be clarified?
5. In Lemma 1, it is stated that g can be approximated by a constant function. It seems more logical that it should be approximated by a quadratic function, is this a typo?
6. Can the authors discuss how the proposed method compares with existing work on direct loss optimization such as [1]?

### Minor comments:
1. To improve clarity, it would be beneficial to summarize the overall computational graph at the beginning: the model "backbone" yields a classifier score, which is used to construct the parameters of the LP, which itself is then solved to compute the metric in a differentiable way. This organization is not clear at the first read.
2. L.104 the second sum for $X_-$ should be over $j$
3. Results on nonnegative matrix factorization are announced in the experiments section, but turn out to be available in the supplementary material only.
4. It is confusing to use he notation $\epsilon$ for both the Linear Program Newton method and the relaxation, since they have different meanings in these two contexts.

### Reference

[1] Song, Yang, Alexander Schwing, and Raquel Urtasun. "Training deep neural networks via direct loss minimization." International Conference on Machine Learning. PMLR, 2016.

----

### Updated Review

Thanks to the authors for their detailed and informative response which addresses all my questions. I am updating my rating from 6 to 7, counting on the authors to update the manuscript so that it has (1) a clearer exposition of the method and (2) a more thorough related work section (including references suggested here and in other reviews).

**Time Spent Reviewing:**

6

---

> ### Author Response · Authors · 2021-08-10
> **Author response for Reviewer mCRV**
>
> Thank you for the time spent in reviewing our paper! We carefully address all your concerns and questions below.
>
> Q1: The experimental evaluation does not seem to provide a timing comparison with the baselines.
>
> A1: We do provide the runtime comparison of our method and the blackbox solver [30] in Fig 3(b) which appears to be the strongest baseline. We are happy to add cross-entropy as well in our final version.
>
> Q2: This work is difficult to follow and sometimes unclear.
>
> A2: Please see our response to reviewer Gi3P where we explain our choice of notations and several unclear places. Some of these were presentation choices to maintain consistency with a number of different papers but we will be glad to modify.
>
> Q3: Why is it stated that the number of constraints $m$  is much larger than the number of variables? It seems to me that in e.g. 2.2 there are about as many variables as constraints
>
> A3: Implicitly, the number of variables is a deterministic function of the minibatch size, say $v(B)$. The number of constraints is determined by the choice of relaxation, $R(v(B))$ (including nonnegativity of slack variables etc.). In our paper, we show that the composition $R \circ v$ is often polynomial in $B$ for many machine learning applications. In practice, the number of constraints is usually more than twice the size of the number of variables due to slack variables and so on. When the number of variables becomes large (and hence the number of constraints), it will become hard to proceed without using SDR or sparse operations, which our modification of the vanilla Newton’s method in (5) and (6) directly enables.
>
>
>
> Q4: Why is epsilon needed in the formulation of 2.2? If the computation of the AUC outputs only binary values, then could not one just use the indicator function  $z_{ij} > 1$ (and "backpropagate" through this operation with 0/1 according to whether the value is 0/1)?
>
> A4: We use epsilon following [4], whose role is to help compute AUC scores (see lines 110-112). It does *not* influence the way we compute gradients. On the other hand, the indicator function itself does not allow the computation of the backward gradient since the gradient is all zero except one point. In practice, the reviewer will agree that most approaches which choose to use an indicator function eventually use a soft version to approximate it (e.g., the logistic function).
>
> Q5: (a) Is it correct to say that at each iteration, the metrics are only computed on the current mini-batch? (this should be made clearer in the paper).
>
> (b) If so, how does the batch-size affect the performance of the algorithm, since the metrics are not decomposable over the samples?
>
> A5: Yes this is correct. At each iteration the metrics are only computed on the current mini-batch (we will make this more explicit in the paper). Empirically, we observe that when the batch-size is too small (smaller than 8, the performance will show a significant drop. When the batch-size is bigger than 32 (we tried 32, 64, 128), we do not see a significant influence of the batch-size on any of the various datasets we performed our experiments on. We also note that Eban et al [9] and others have reported success with using per-example (or mini-batch) quantities for specific non-decomposable measures.
>
> Q6: L.177 the paragraph title is "Why is Newton’s method applicable for minibatches?" but the content does not seem to address this question, can this be clarified?
>
> A6: This is a great question! For the LP examples shown in the paper, the exterior penalty function has many favorable properties, importantly, the primal-dual relationship. First, our results in Lemma 1 and Theorem 1 trivially implies that we do *not* require any complex line search procedures for solving LPs (defined by individual minibatches) using the Newton’s method, and hence it is indeed applicable in each iteration. Moreover, for backpropagation purposes, we would like to use an efficient method such as Newton’s method since unrolling can be memory intensive. Our discussion in this section provides justification as to why it is safe to use Newton’s method for the examples shown in the paper since convergence properties of Newton’s method for machine learning purposes remains an active area of research. Our discussion in this paragraph completely clarifies both the difficulties faced when using Newton’s method for nonsmooth problems such as LPs and so it is indeed applicable in training schemes that use variable minibatch sizes (where line search would be quite problematic).
>
> Q7: In Lemma 1, it is stated that g can be approximated by a constant function. It seems more logical that it should be approximated by a quadratic function, is this a typo?
>
> A7: Yes, we meant it to be a quadratic function. Thanks for catching the typo.
>
> Q8: Can the authors discuss how the proposed method compares with existing work on direct loss optimization such as [1]? ([1] Song, Yang, Alexander Schwing, and Raquel Urtasun. "Training deep neural networks via direct loss minimization." International Conference on Machine Learning. PMLR, 2016.)
>
> A8: At a high level, the work on direct loss optimization [1] is similar in spirit with the works on blackbox solvers in the sense that they both solve an argmax/argmin problem in a non-differentiable way (often require solving LP multiple times) and then carefully designed procedures for the backward gradient that often require at least *one more* LP solve, albeit a different LP. Furthermore, [1] uses the finite difference method to approximate the gradient which often introduces numerical stability issues in practice. To summarize, there are two direct benefits of using our method over [1]; (i) Our proposed method directly solves the target problem (e.g., AUC) in a differentiable way and does not need extra approximations to estimate the backward gradient, and (ii) our method does not suffer from the possible numerical error brought by the finite difference method used in [1] which essentially amounts to tuning more accuracy parameters (per feature dimension).
>
> Q9: To improve clarity, it would be beneficial to summarize the overall computational graph at the beginning.
>
> A9: We agree that this is a good suggestion. We will include a summary for easier understanding.
>
> Please let us know if there are any other questions we can answer.

---

> ### Author Response · Authors · 2021-09-01
> **Thanks to Reviewer mCRV**
>
> We thank Reviewer mCRV for reading our response and the updated review. We will make the changes about the clarifications and add a more thorough related work section as mentioned in our previous response.

---

### Official Review · Reviewer_TyHW · 2021-07-16

**Rating:** 7
**Confidence:** 3

**Summary:**

This paper present a novel approach to perform direct optimization of non-decomposable objective functions (such as AUC or F-measure) in the context of Deep Learning.
They propose a general LP-based framework describing these objective functions and provide a general algorithm to solve this problem (and get the corresponding gradients).
Finally, this new approach is compared to other methods trying to optimize these objectives directly and surrogate loss functions that are commonly used in practice.

**Limitations And Societal Impact:**

The authors clearly address the limitations of the method in a clear paragraph.


**Main Review:**

I think the paper is well written and strike a good balance of technical details and high level intuition. And clearly states which results are included or left in the supplementary materials.

From a high level:
- While stochastic training is omnipresent and explicitly mentioned l33 and l177, there is no discussion in this paper that discuss the impact of the impossibility of computing the exact objective function on the whole dataset. A lot of these more principled methods usually struggle to reconcile this idea that the objective itself cannot be computed to begin with (let alone quickly or be optimized). Could you clarify how the method presented here is positioned with respect to that? In particular for the LP formulations in Section 2.
- l41 mentions that the LP that is being solved is a relaxation in  these cases, it would be nice to have some details on how good this relaxation is? And if it could be interesting to get a more tight one?
- The limited number of iteration used for this algorithm also remind me of other "optimization as layer" work done on CRFs. Where a limited numbers of optimization steps are done and provide most of the benefit. Maybe citation of such work would give a nice backing for this choice here.
- l205-207 such hyperparameter can have a big impact in practice on the result and cross validating them is a significant endeavor (especially when different part of your network have such parameters). Do you have any experimental result on this sensitivity? And how crucial is the lambda tuning?
- l214 do you have any result on data-driven SDR?
- The absence of baseline trying to optimize the multiclass AUC is surprising. Could you detail why?

Smaller details:
- Thanks for including a notations section!
- l165 "exterior penalty in (3)" is not defined
- l168-169: I am not sure what this sentence is showing? In particular, you need to be able to evaluate your function efficiently to be able to use it. I don't see why AD is used as an argument here.
- l189 the value given in parenthesis is a pair of values?
- l225 the sentence is not english. More generally, a few sentence in the paper would gain at being shorter.
- l234, why do you redifine Q = H tilde ? Why not just keep using H tilde?
- l252 using -> during
- l257 the discussion around Danskin's theorem is surprising here as it is not mentioned anywhere else in the paper.
- Fig 2 is never linked from the text, same for Fig 3a as far as I can see
- l316 formatting is weight. It could be better aligned to avoid the single dangling line
- l324 which results?
- Table 3: "N" should be defined in the caption.



**Time Spent Reviewing:**

2

---

> ### Author Response · Authors · 2021-08-10
> **Author response for Reviewer TyHW**
>
> Thank you for the time spent in reviewing our paper! We carefully address all your concerns and questions below.
>
> Q1. Could you clarify how the method presented here is positioned with respect to the fact that the objective on the whole dataset may be impossible to compute to begin with?
>
> A1: This is a great question! From the computational perspective, yes, the gradient is usually impossible to compute exactly (hence optimize) for the whole dataset especially when the objective function is an expectation over a random variable (data) with infinite support, common in machine learning setups. This is one reason why we explored the use of SDR as the theoretical framework for sparse operations for efficiency gains in (5) and (6). Empirically, in lines 354-356, we note that a mini-batch size of 200 is the limit for the AUC computation on a 2080TI GPU *without* using sparse operations. Computing (and optimizing) AUC for the whole dataset is clearly intractable.
>
> Indeed, AUC is a functional on the space of probability measures and as such optimizing it is a generic nonconvex discrete optimization problem. Our results correspond to the setting where we are given a finite set of samples, with the goal of obtaining a nontrivial estimator for the gradient -- nontrivial because the problem is discontinuous (piecewise constant in many cases), and the gradient vanishes almost everywhere. From the statistical perspective, our results (including formulation and backpropagation procedures) here can be interpreted as a (consistent) finite estimator of the gradient for AUC type functionals. In general, since any statistical estimator can only be approximated to a certain probability (by definition), this is true for our estimator as well. On a related note, a number of works including Eban et al [9] tackle non-decomposable measures such as AUROC via scalably optimizing bounds on per-example quantities.
>
> Q2: Line 41 mentions that the LP that is being solved is a relaxation in these cases, it would be nice to have some details on how good this relaxation is?
>
> A2: Empirically we found that the relative error between the true AUC score (given by correct IP solve) and the score estimated using our LP formulation and proposed solver is less than 1 percent during training, on the Cat&Dog dataset as an example although there could be datasets or situations where the IP/LP gap is worse.
>
> From the theoretical perspective, we found that a unified theoretical analysis of the integrality gap for general purpose Newton iterations (or an alternative algorithm) for all of the different objective functions we have covered does not appear to be within reach. Generic randomized rounding type schemes are indeed applicable and it may be possible to derive (dimension dependent) approximation ratios/integrality gaps, although we are not aware of tightness of a practical rounding scheme that could work for all our objectives.  More generally, one option is to consider LP/SDP hierarchies for provable guarantees with quasipolynomial runtime requirements, whenever certain moment conditions on the features are satisfied. However, these results are too coarse for our purposes since we solve the relaxation in each iteration, and so these results when extrapolated across iterations are too pessimistic -- success probability rapidly diminishes as iterations increase.
>
> On the practical side, our experiments with deep networks clearly indicate that approximate gradients provided by our continuous relaxations are sufficient, and moreover, beneficial for predictive purposes.
>
> The reviewer will appreciate that from the theoretical standpoint, dependency between iterates is the biggest challenge, that is, analyzing the success probability for dependent variables (iterates) is a highly nontrivial task, if not impossible, especially in high dimensions considered here.
>
> Q3: The limited number of iteration used for this algorithm also remind me of other "optimization as layer" work done on CRFs. Maybe cite some of that?
>
> A3: Yes this is a good observation. The use of a limited number of iterations indeed also appears in the segmentation literature where we would use CRFs with a few iterations after the DNN module (Deeplab: Semantic image segmentation, Chen et. al., 2017), and in message passing neural networks where one assumes that several iterations of message passing can provide most benefits (Neural message passing for quantum chemistry, Justin et. al., 2017). More recently (Understanding Deep Architectures with Reasoning Layer, Chen et. al., 2020), optimization algorithms have been successfully used to design such probabilistic reasoning type modules within deep networks -- energy functions in CRF models can simply be used as objective functions in optimization layers. We will be happy to include additional discussion of these links. Thank you.
>
> Q4: l205-207, how crucial is the tuning of $\lambda$?
>
> A4: Very interesting question which highlights a key strength. Note that (3) is a piecewise quadratic function and so the value of the optimal lambda depends on the piece that contains the optimal solution. Our Lemma 1 shows that when the columns of $A$ are in general position, it is almost certainly guaranteed that the objective function is continuous on any neighborhood of the optimal solution -- this means that there is a range of lambda for which the algorithm converges to the same optimal solution!
>
> Empirically, we tried $\lambda=0.1, 0.2, 0.5, 1$ and *all* choices give estimates of AUC score to the same accuracy level (within 1 percent relative error w.r.t. the true AUC score, using Cat&Dog dataset as an example).
>
> Q5: Line 214: do you have any result on data-driven SDR?
>
> A5: Yes, please see Fig 3(c). Here we show the relationship (tradeoff) between the reduced size and the final accuracy.
>
> Q6: The absence of a baseline trying to optimize the multiclass AUC is surprising. Could you detail why?
>
> A6: Thank you for bringing up this question. To our knowledge and based on feedback/discussions with some groups active in this area, we are not aware of any other results which differentiably optimize *multiclass* AUC. We welcome suggestions on any specific baselines that could further strengthen our experiments.
>
> We hope the reviewer will agree that the PPD algorithm given in [19] is designed for *binary* 2-class AUC. The authors did not provide an algorithm for multiclass AUC in that paper or conduct experiments for multiclass AUC. As a result, we decided to use Cross-Entropy as a baseline, which is a reasonable baseline because, in the ideal setting, it will guide the network to the same optimal point as our AUC loss (the optimal point is where the model achieves 100 percent classification accuracy). We can also see from Table 1 that the Cross-Entropy has close performance to ours and PPD algorithm in some cases where the dataset is nearly balanced.
>
> Q7: other details/typos
>
> A7: We thank the reviewer for pointing these out and we will make corresponding modifications in our final version.
>
> Please let us know if there are any other questions we can answer.

---

### Decision · Program_Chairs · 2021-09-27

**Decision:**

Accept (Poster)

**Comment:**

Thank you for your submission to NeurIPS.  The reviewers and I are in agreement that the proposed work presents some substantial advances to the topic of optimizing non-decomposable metrics, in this case using a nice application of unrolled smoothed LP solving.  The reviewers provided several comments, and in some cases have already improved their scores after rebuttal.  I would only suggest that the authors be sure to make these modifications, but I have no concerns, as these are largely easy to make.  I'm happy to recommend the paper for acceptance.